# Human-Induced Neural and Mesenchymal Stem Cell Therapy Combined with a Curcumin Nanoconjugate as a Spinal Cord Injury Treatment

**DOI:** 10.3390/ijms22115966

**Published:** 2021-05-31

**Authors:** Pablo Bonilla, Joaquim Hernandez, Esther Giraldo, Miguel A. González-Pérez, Ana Alastrue-Agudo, Hoda Elkhenany, María J. Vicent, Xavier Navarro, Michael Edel, Victoria Moreno-Manzano

**Affiliations:** 1Neuronal and Tissue Regeneration Laboratory, Centro de Investigación Príncipe Felipe, 46012 Valencia, Spain; pbonilla@cipf.es (P.B.); egiraldo@cipf.es (E.G.); miguelangelgozalezperez14@gmail.com (M.A.G.-P.); aalastrue@cipf.es (A.A.-A.); hoda.atef@alexu.edu.eg (H.E.); 2Neuroplasticity and Regeneration Group, Department Cell Biology, Physiology and Immunology, Institute of Neurosciences, Universitat Autònoma de Barcelona and CIBERNED, 08193 Bellaterra, Spain; Joaquim.Hernandez@uab.cat (J.H.); Xavier.Navarro@uab.cat (X.N.); 3Department of Biotechnology, Universitat Politècnica de València, 46022 Valencia, Spain; 4Department of Surgery, Faculty of Veterinary Medicine, Alexandria University, Alexandria 22785, Egypt; 5Polymer Therapeutics Laboratory, Centro de Investigación Príncipe Felipe, 46012 Valencia, Spain; mjvicent@cipf.es; 6Laboratory of Regenerative Medicine, Institut Barraquer, 08021 Barcelona, Spain; edel.michael@gmail.com

**Keywords:** spinal cord injury, stem cells, curcumin, neuroprotection, polymer–drug conjugate

## Abstract

We currently lack effective treatments for the devastating loss of neural function associated with spinal cord injury (SCI). In this study, we evaluated a combination therapy comprising human neural stem cells derived from induced pluripotent stem cells (iPSC-NSC), human mesenchymal stem cells (MSC), and a pH-responsive polyacetal–curcumin nanoconjugate (PA-C) that allows the sustained release of curcumin. In vitro analysis demonstrated that PA-C treatment protected iPSC-NSC from oxidative damage in vitro, while MSC co-culture prevented lipopolysaccharide-induced activation of nuclear factor-κB (NF-κB) in iPSC-NSC. Then, we evaluated the combination of PA-C delivery into the intrathecal space in a rat model of contusive SCI with stem cell transplantation. While we failed to observe significant improvements in locomotor function (BBB scale) in treated animals, histological analysis revealed that PA-C-treated or PA-C and iPSC-NSC + MSC-treated animals displayed significantly smaller scars, while PA-C and iPSC-NSC + MSC treatment induced the preservation of β-III Tubulin-positive axons. iPSC-NSC + MSC transplantation fostered the preservation of motoneurons and myelinated tracts, while PA-C treatment polarized microglia into an anti-inflammatory phenotype. Overall, the combination of stem cell transplantation and PA-C treatment confers higher neuroprotective effects compared to individual treatments.

## 1. Introduction

The physiopathology of spinal cord injury (SCI) involves the disruption of spinal pathways and a primary succession of processes initiated immediately after trauma that prompt rapid and massive cell death within the nervous tissue and the concomitant invasion of ectopic immune and fibrotic cells [1]. During a secondary injury phase, neuronal death expands to neighboring segments, and neuroinflammation and the associated non-permissive microenvironment [2] limit the capacity of adult neurons to spontaneously regenerate axons [3]. Therefore, a multifactorial approach to SCI treatment addressing the broad range of pathological factors involved may have better success than conventional monotherapeutic advances that focus on a single factor.

Stem cell transplantation, particularly the use of neural stem cells (NSC) [4], represents a potentially effective treatment approach for SCI. Pre-clinical assays have demonstrated that NSC can replace lost host cells, bridge lesions [5], recover myelin sheaths to reconstitute neuronal circuitry and lost connectivity [6], provide trophic support [7], and modulate neuroinflammation [8]. The US Food and Drug Administration (FDA) approved an immortalized NSC line (NSI-566), derived from human early fetal spinal cord tissue, to treat chronic SCI in phase I clinical trials reporting no serious adverse effects [9]. The administration of NSI-566 in several pre-clinical models of SCI (rodents [10] and non-human primates [11]) prompted significant improvements to neurological function and suppressed spasticity by supporting extensive axonal sprouting and the development of synaptic contacts with host neurons. However, NSI-566 transplantation requires immunosuppression, which is a disadvantage that could be avoided via the use of derivatives of autologous or immune-compatible induced pluripotent stem cells (iPSC).

The induced differentiation of human iPSC into NSC (iPSC-NSC) provides a proliferative and broadly expandable in vitro cell source with glial [12] and neuronal differentiation potential [13,14]. In 2019, Okano et al. described the first licensed trial for the clinical evaluation of iPSC-NSC in chronic SCI treatment in Japan [15]. Okano’s group had previously employed rodent [13] and primate [16] models to demonstrate how iPSC-NSC significantly improved locomotion after severe to moderate traumatic SCI [17]. Despite the beneficial effects of iPSC-NSC, including their autologous nature that avoids immune rejection [18], their potential tumorigenicity remains a significant impediment to their clinical use [19]. In this regard, we recently generated genetically stable human iPSC using a modified reprogramming procedure employing the transfection of synthetic mRNAs coding for CYCLIN D1 and the OSKL reprogramming factors (OCT3/4, Sex determining Region Y-box 2 (SOX2), Kruppel Like Factor 4 (KLF4), homolog lin-28 (LIN28)), thereby avoiding the use of C-MYC [20], under clinically compatible conditions. iPSC-NSC subsequently derived from these newly generated iPSC displayed reduced genetic instability, reduced cell proliferation in teratoma assays, and efficient survival, engraftment, and differentiation in a hostile SCI microenvironment [20].

Numerous related approaches have also explored the safety and efficacy of mesenchymal stem cell (MSC)-based therapies for SCI [21,22,23,24,25] and highlighted MSC transplantation as an interesting means to locomotor recovery in animal models [26]. Our recent studies indicated that the transplantation of allogeneic MSC combined with immunosuppression supports the survival of engrafted cells, which improves functional and morphological outcomes after SCI [27]. The therapeutic benefits of MSC primarily relate to the secretion of paracrine acting factors [28] that provide neural support and promote remyelination [29]. In addition, MSC also display immunomodulatory [30], anti-apoptotic [31], and angiogenic [32] abilities but lack tumor-initiating potential [33]. Furthermore, given their accessibility and feasible isolation, MSC can be used in an autologous manner.

Given the pathological heterogeneity of SCI, multifaceted combinatorial approaches have been developed that induce improvements regarding cell grafting and survival [11,34], neural differentiation [35], axonal regeneration [35], and the prevention of secondary damage [36]. We previously described a successful combination therapy comprising NSC transplantation and treatment with a water-soluble pH-responsive polyacetal–curcumin nanomedicine incorporating curcumin in the polyacetal mainchain (PA-Curcumin or PA-C), which supports the local sustained release of the active compound curcumin [37]. This synergistic combination led to increased levels of neuroprotection and induced functional recovery in a severe model of chronic SCI in adult rats [37].

In the current study, we report on the development of an enhanced synergistic combination therapy comprising PA-C and iPSC-NSC + MSC treatment as a treatment option for severe sub-acute traumatic SCI. This advanced approach combines the induction of neuroprotection and neuroregeneration to target the secondary pathological mechanisms that occur after SCI.

## 2. Results

### 2.1. Cultured iPSC-NSC Express Neuronal, Astroglial, and Oligodendrocyte-Specific Lineage Markers

We first characterized the heterogeneous iPSC-NSC derived from cyclin D-reprogrammed iPSC [20]. Immunofluorescent evaluation of the three neural lineages performed under proliferative culture conditions (representative images shown in Figure 1A) demonstrated a significant percentage of cells positive for early neural precursor markers, including SOX2 (75.39 ± 3.24% of total cells), NESTIN (93.05 ± 2.69% of total cells), Paired box protein (PAX6) (45.98 ± 5.01% of total cells), and neurogenic locus notch homolog protein 1 (Notch1) (39.76 ± 5.63% of total cells), which confirmed their neural lineage [38,39] (Figure 1B). Furthermore, we discovered that around one-third of iPSC-NSC expressed Forkhead Box J1 (FOXJ1), which is a marker for ciliated cells that include the neuroepithelial cells lining the ependymal canal of the spinal cord [40]. iPSC-NSC also expressed markers of neurons (β-III Tubulin—37.05 ± 4.06% of total cells, and doublecortin (DCX)—54.8 ± 4.23% of total cells), glia (glial fibrillary acidic protein (GFAP)—48.35 ± 5.97% of total cells), and oligodendrocytes (O4—39.97 ± 8.01% of total cells) (Figure 1A,B). We previously showed comparable neuronal and glial cell identity in iPSC-NSC cultures upon neural induction medium conditions, with about 60% of β-III Tubulin-positive cells and about 34% of GFAP-positive cells [14].

### 2.2. PA-C Treatment Increases iPSC-NSC Viability, Enhances Neurite Elongation, but Fails to Induce Neural and Glial Differentiation

We previously described that polyacetal–curcumin conjugate (PA-C, curcumin loading 3.8% *w*/*v*) displayed significantly lower cytotoxicity than free curcumin (C) in primary cultures of adult rat spinal cord NSC [37]. To elucidate whether PA-C influences the viability of iPSC-NSC, we performed an MTS (3-(4,5-dimethylthiazol-2-yl)-5-(3-carboxymethoxyphenyl)-2-(4-sulfophenyl)-2H-tetrazolium) cell viability assay at 24 h after treatment with increasing concentrations (up to 20 µM) of PA-C at curcumin equivalent concentrations. We discovered that the evaluated concentrations of PA-C failed to induce cytotoxicity in iPSC-NSC; moreover, PA-C treatment significantly increased iPSC-NSC metabolic activity in a dose-dependent manner at concentrations higher than 10 µM (Figure 2A).

Next, we studied whether a 24 h treatment with 10 µM PA-C influenced iPSC-NSC fate determination. Staining for cells positive for GFAP or β-III-Tubulin indicated that PA-C treatment failed to alter the proportion of astrocytes and neurons, respectively, compared to control vehicle-treated iPSC-NSC (Figure 2B). We also analyzed gene expression profiles using an RT2 Profiler PCR Array that includes a set of genes related to neural differentiation such as Brain-Derived Neurotrophic Factor (BDNF), Oligodendrocyte transcription factor (Olig2), SOX2, Achaete-Scute Family BHLH Transcription Factor 1 (ASCL1), Dopamine receptor D2 (DRD2), and apoptosis-related genes such as Adenosine A1 receptor (ADORA1) and BCL2 (Appendix A). PA-C treatment failed to influence the expression of genes involved in apoptosis, which agrees with the results of the cytotoxicity assays (Figure 2A). Furthermore, from this select group of genes, only DRD2 expression became significantly downregulated after treatment with PA-C (* *p* < 0.05 compared to control vehicle-treated iPSC-NSC; Appendix A), indicating that PA-C treatment did not significantly induce neural differentiation of iPSC-NSC. Previous studies established that PA-C treatment induced neurite outgrowth [37]. While the incubation of iPSC-NSC with 10 µM PA-C for 24 h failed to induce significant neural differentiation, we found that PA-C treatment significantly induced neurite elongation in β-III Tubulin-positive cells (Figure 2C,D). We also evaluated the functional relevance of induced neurite elongation by PA-C treatment in an inhibitory environment by treating iPSC-NSC with lysophosphatidic acid (LPA), which activates the Rho/ROCK pathway and induces growth cone retraction and neurite collapse [41,42]. We employed free fasudil (Fas) [43], a well-known Rho kinase inhibitor, and a nanoconjugated form of fasudil (PGA-SS-Fas) [44] as positive controls for neurite elongation. In a similar manner to curcumin, we previously reported that polymer conjugation enhanced the stability of fasudil and supported sustained release, which improved the neuroprotective and regenerative activity of fasudil in SCI models [44]. While treatment with PGA-SS-Fas or Fas (Figure 2E,F) permitted a significant increase in axonal elongation in the presence of LPA, we found no differences following the treatment of iPSC-NSC with PA-C (Figure 2D) or C (Figure 2E,F). Representative images of β-III-Tubulin immunostaining in the absence of LPA treatment show neurite elongation following PA-C treatment (Figure 2C), abundant and long neural extensions following PGA-SS-Fas treatment (Figure 2E,F), and the absence of morphological changes in PA-C-treated iPSC-NSC in the presence of LPA in comparison with vehicle-treated control iPSC-NSC (Figure 2E,F).

### 2.3. PA-C Protects iPSC-NSC against Hydrogen Peroxide-Induced Toxicity

The excessive production and release of reactive oxygen species following SCI contribute to the secondary injury phase by exacerbating acute damage to spinal cord-resident cells [45,46] and generating a hostile microenvironment for cell-based therapies. Antioxidant treatments have emerged as an effective approach to ameliorate cell death/secondary damage. We evaluated PA-C pre-treatment (5, 10, 12.5, 15, 17.5, or 20 µM) in the prevention of oxidative cell damage following the exposure of iPSC-NSC to high doses of hydrogen peroxide (H_2_O_2_) (75 or 100 µM) for 24 h. We determined H_2_O_2_-induced cytotoxicity by MTS viability assay, discovering that PA-C treatment in the range of 5 to 15 µM prevented 75 µM H_2_O_2_-induced cytotoxicity closed to the control non-treated condition. Representative phase-contrast images evidenced a cell survival-inducing effect of 10 µM of PA-C in the presence of 75 µM H_2_O_2_ (Figure 3A); however, PA-C pre-treatment failed to protect iPSC-NSC treated with 100 µM H_2_O_2_ (Appendix A).

Due to the heterogeneous nature of cultured iPSC-NSC (Figure 1), we evaluated the potential neuronal/glial selectivity of PA-C treatment using a 30 min 10 µM PA-C pre-treatment (or vehicle treatment in control) followed by a 24 h treatment of 75 µM H_2_O_2_. We discovered that H_2_O_2_ significantly inhibited the survival of neuronal progenitors, as shown by a decrease in the levels of β-III Tubulin-positive neural cells, but it had no effect on glial progenitor survival, as shown by the unchanging proportion of GFAP-positive glial cells (Figure 3B shows presentative images and the quantification of β-III Tubulin- and GFAP-positive cells). Furthermore, 10 µm PA-C pre-treatment significantly protected neuronal progenitor cells from H_2_O_2_-induced cytotoxicity (Figure 3B).

### 2.4. PA-C Treatment and MSC Prevent NF-κB Translocation to the Nucleus of LPS-Treated iPSC-NSC

We assessed the anti-inflammatory effect of a 24 h pre-treatment of 10 µM PA-C on iPSC-NSC co-cultured with MSC subsequently treated with 1 µg/mL LPS (lipopolysaccharides, a bacterial endotoxin) for 24 h to induce the nuclear translocation and activation of NF-κB, a critical mediator of pro-inflammation following SCI [47]. As shown by immunostaining images (Figure 4A) and quantification (Figure 4B), PA-C treatment did not significantly affect NF-κB translocation in iPSC-NSC monocultures or iPSC-NSC + MSC co-cultures. While LPS-induced inflammatory stress significantly increased nuclear NF-κB translocation (Figure 4A; white arrows) and activation in iPSC-NSC monoculture, LPS treatment in iPSC-NSC + MSC co-culture did not induce NF-κB translocation. Furthermore, pre-treatment with PA-C significantly reduced LPS-induced NF-κB translocation in iPSC-NSC monocultures. However, no additional effect of PA-C is observed in the co-cultures, as MSCs alone inhibit NF-κB translocation. (Figure 4A; yellow arrows). Overall, these findings provide evidence that PA-C pre-treatment and MSC co-culture inhibit the activity of inflammatory mediator NF-κB in iPSC-NSC cultured under pro-inflammatory stimulus conditions.

### 2.5. Functional Locomotor Recovery after SCI in Response to Individual Treatments and Combination Therapies

In vivo comparisons employed adult female rats subjected to traumatic SCI by severe contusion of 200 kdynes at the thoracic T8 level. We subjected all animals to a second surgery after one week to transplant iPSC-NSC, MSC, or iPSC-NSC and MSC. We evaluated PA-C treatment by intrathecal delivery for one week in control SCI rats that received no cell therapies and experimental rats that received the iPSC-NSC + MSC treatment. We evaluated rats using the Basso, Beattie, and Bresnahan (BBB) open-field locomotor scale test [48] for nine weeks after SCI to assess functional recovery (Figure 5A).

Two weeks after SCI (one week after cell transplantation), we found statistical differences between the iPSC-NSC and the PA-C treated group (* *p* < 0.05) and between the iPSC-NSC and the combinatory cell transplant (iPSC-NSC + MSC; # *p* < 0.01). However, we found no significant differences in any of the experimental groups compared to control SCI rats.

Nevertheless, BBB analysis after treatment with iPSC-NSC or MSC led to better overall functionality resulting in faster improvement after the first week of treatment with higher slopes in the BBB curves (Figure 5A, green and dark blue lines); however, animals receiving cell transplantation (iPSC-NSC, MSC, or iPSC-NSC and MSC) reached a plateau in their BBB curves four weeks after injury. PA-C-treated animals displayed continuous improvements over time (Figure 5A, orange line).

To better visualize these improvements, we subdivided the BBB score [48] (Figure 5B) into three different ranges: from 0 to 8 (indicating locomotion without supporting body weight), from 9 to 10 (indicating steps supporting body weight without co-ordination), and from 11 to 21 (indicating coordinated stepping). This separate analysis indicates the percentage of animals receiving each treatment that reach the locomotion skills established at each BBB range. As shown in Figure 5B, almost 80% of animals treated with MSC or PA-C reached a BBB range of 11–21. The number of animals receiving iPSC-NSC, MSC, or PA-C treatment reaching the highest BBB range was more significant than the number of animals receiving the iPSC-NSC + MSC and iPSC-NSC + MSC + PA-C treatments (Figure 5B, lower panel). The control SCI group had a higher percentage of animals in the 0–8 range, with only 50% reaching the 11–21 range.

Locomotor function analysis using Catwalk Gait at nine weeks post-SCI (Appendix A) highlighted significant results in iPSC-NSC-treated SCI rats. These animals displayed an increased maximum contact in seconds (registered maximum contact time within the hind paws during free and straight walking) compared to control SCI animals.

### 2.6. PA-C and PA-C Combined with iPSC-NSC + MSC Transplantation Preserve β-III-Tubulin Positive Fibers, Limit Inhibitory Scar Size, and Increase Functional Synapse Number after SCI

We analyzed longitudinal spinal cord sections, including the epicenter of the injury and rostral and caudal segments to the injury, by double immunostaining with β-III-Tubulin (representative images shown in Figure 6A; left panels, green) and GFAP (right panels, red) to quantify the nerve fiber preservation. The quantification of β-III-Tubulin staining (Figure 6B; top panel) revealed a significantly larger positive area present in animals receiving iPSC-NSC + MSC + PA-C treatment than control animals or those receiving iPSC-NSC and MSC but not in comparison with PA-C treatment. PA-C-treatment prompted a significantly larger β-III Tubulin-positive area than iPSC-NSC or MSC treatments. Of note, we observed a significant reduction in the number of neuronal fibers following iPSC-NSC or MSC treatment compared to the control or iPSC-NSC + MSC + PA-C treatment.

We also evaluated the size of the inhibitory scar by quantifying the lack of GFAP staining, expressing this value as a percentage of the total quantified tissue area (Figure 6A-right panel, dashed line; Figure 6B-bottom panel). PA-C and iPSC-NSC + MSC + PA-C treatments significantly reduced scar area compared to other treatments. iPSC-NSC treatment resulted in an increased scar area compared to control; however, iPSC-NSC + MSC or MSC treatments failed to induce any significant alterations.

To evaluate the preservation of functional synapses, we quantified the synaptic bottoms at neuronal somas by co-localizing synaptophysin and NeuN for those treatments providing better outcomes in the previous neuronal fiber preservation analysis (Figure 6C shows representative images of each of the tested staining’s from the control group). Analysis of preserved NeuN-positive cells, expressed as the percentage of NeuN-positive area, demonstrated a significant increase following PA-C treatment compared to control (Figure 6D; left panel). Furthermore, the quantification of co-localization (synaptophysin/NeuN+) normalized to the total NeuN-positive area demonstrated a significant increase following PA-C or iPSC-NSC + MSC + PA-C treatment when compared to control (Figure 6D; right panel).

### 2.7. PA-C Combined with iPSC-NSC + MSC Transplantation Induces White Matter Sparing after SCI

We determined white matter sparing by Luxol fast blue (LFB) staining—LFB binds to the myelin sheath lipoproteins and allows quantification of the remaining myelinated areas [49]. LFB analysis (Figure 7A,B) demonstrated that iPSC-NSC and MSC treatments increased spinal cord white matter sparing compared to control and PA-C treatment. Furthermore, iPSC-NSC + MSC + PA-C treatment also increased white matter preservation compared to control and PA-C treatment.

We also quantified the number of motoneurons (MN) at the epicenter of the lesion (T8), both rostrally (T7) and caudally (T9) (expressed as MN/mm^2^), as indicated in the scheme in Appendix A. The segmented analysis (Appendix A) demonstrated that iPSC-NSC, MSC, or iPSC-NSC + MSC treatment significantly prevented motoneuron loss compared to the control, PA-C treatment, and iPSC-NSC + MSC + PA-C treatment at the rostral segments (T7). Quantification of motoneurons in caudal segments to the lesion failed to demonstrate significant differences in preservation for any treatment compared to control; however, iPSC-NSC, MSC, or iPSC-NSC + MSC treatment led to more significant levels of motoneuron preservation compared to PA-C treatment. Furthermore, iPSC-NSC treatment increased motoneuron preservation compared to iPSC-NSC + MSC + PA-C treatment (Appendix A).

### 2.8. PA-C and PA-C Combined with iPSC-NSC + MSC Transplantation Promotes Microglia Polarization toward an Anti-Inflammatory Profile 

We Next Evaluated the Potential in Vivo Anti-Inflammatory Activity of PA-C or iPSC-NSC + MSC + PA-C by Studying the Presence of Activated Microglia with an Anti-Inflammatory Profile by the Co-Localization of IBA1 Expression (Microglia Marker) and Arginase-1 (Arg1) (Figure 8A,B). After Quantifying the Percentage of Cells Expressing IBA1, Arg1, or IBA1 and Arg1 (Figure 8C), We Found that both Treatments (PA-C or iPSC-NSC + MSC + PA-C) Increased the Presence of Anti-Inflammatory Microglia (IBA1 + Arg1) in the Spinal Cord Nine Weeks Post-SCI.

## 3. Discussion

Combination therapies have emerged as a promising strategy to inhibit or slow the highly complex and heterogeneous cascade of pathological events following SCI. Combinations of stem cell therapies, including MSC and NSC, have been reported for SCI treatment of SCI, although with contradictory results [50,51,52]. Park et al. demonstrated that subacute transplant of human NSC led to significant functional recovery; in contrast, human MSC failed to provide any effect alone or in combination with NSC [50]. In another study, the co-transplantation of MSC and NSC enhanced motor function and axon density surrounding the lesion; however, stem cell co-transplantation failed to prompt white matter sparing and induced tumor formation [51]. Finally, Sun et al. demonstrated that the sub-acute co-transplantation of human NSC and MSC enhanced stem cell survival compared to individual treatments, increased the number of myelinated axons, and improved locomotor function [52]. The use of MSC secretome, alone or in combination with NSC, would provide a successful alternative to cell therapy, since researchers have ascribed paracrine effects to exosomes in the treatment of spinal cord injuries including anti-inflammatory [53], angiogenic [54] and regenerative effects [55], as well as functional recovery [56].

We previously reported that neonatal rat NPCs transplantation combined with local delivery of PA-C, but not the individual treatments, significantly rescued voluntary locomotion in severe and chronic SCI models, increased neuronal preservation, and reduced scarring at the injury site [37,57]. This study aimed to use a clinically relevant combination therapy approach comprising the transplantation of iPSC-NSC and MSC and PA-C treatment. We discovered that this combination therapy confers certain benefits compared with the single treatments, such as preserving neuronal fibers and reducing scar tissue. Unfortunately, these significant improvements failed to translate into functional improvements.

iPSC-NSC tolerated PA-C well in vitro, inducing cell proliferation and neurite elongation, as had been previously demonstrated in rat NSC by Rho/Rock kinase inhibition [37]. Furthermore, curcumin also displayed neurite elongation inducing activity in mouse neural crest-derived cells through inhibition of the proteasome [58], and in cortical neurons, by the concentration-dependent regulation of the Reggie-1 and ERK1/2 pathways [59]. In our in vitro model, PA-C induced neurite outgrowth in iPSC-NSC under proliferative conditions; however, PA-C failed to counteract LPA-induced neurite retraction or induce in vitro neurogenesis of iPSC-NSC, as had been previously described in embryonal carcinoma stem cells through the activation of autophagy [60].

After the primary trauma, secondary events cause mitochondrial dysfunction and augment reactive oxygen species formation, leading to cell death [46] and neuronal damage [45,61]. We hypothesized that PA-C pre-treatment might protect iPSC-NSC exposed to cytotoxic doses of hydrogen peroxide, given the previously attributed antioxidant and immunomodulatory properties of curcumin [62]. Overall, PA-C counteracted the damaging effects of hydrogen peroxide and increased iPSC-NSC viability, which is a finding supported by a previous study of curcumin treatment in rat NSC [63].

Given the fundamental nature of NF-κB signaling pathway activation to neuroinflammation and SCI pathophysiology [64,65,66], the modulation of NF-κB signaling could ameliorate inflammation, reduce the impact of the secondary damage stage, and preserve neuronal function. A previous study reported that curcumin reduced neuroinflammation after SCI by specifically suppressing the TLR4/NF-κB signaling pathway [67]. In this study, we established of the effect of PA-C on the inflammatory mediators by demonstrating that this pH-responsive nanoconjugate inhibited the LPS-mediated induction of NF-κB signaling and translocation to the nucleus in iPSC-NSC. We also established the potent immunomodulatory properties of MSC in vitro by showing how co-culture inhibited NF-κB activation in iPSC-NSC. Gene profiling analysis revealed a significant downregulation of DRD2 in iPSC-NSC after PA-C S treatment in vitro. As H_2_O_2_ exposure up-regulates DRD2 in the human SH-SY5Y neuroblastoma cell line through NF-κB nuclear translocation [68], PA-C treatment may downregulate DRD2 expression by inhibiting the NF-κB signaling pathway.

The combination of cellular therapies with or without the addition of PA-C failed to significantly improve locomotor skills after traumatic SCI; however, we did observe noticeable differences in individual parameters. Following the subdivision of the BBB scale, we observed a more rapid recovery following iPSC-NSC or MSC treatments; however, animals soon reached a plateau after the initial functional recovery with regard to functional improvements. Interestingly, animals treated with PA-C alone and PA-C combined with MSC and iPSC-NSC displayed non-significant but increasing improvements until the experimental endpoint. The different severity of the injury may account for the inconsistent result compared to previous co-transplant studies, in which combination treatments enhanced locomotor recovery [52]. Despite the absence of significant functional locomotion recovery, the histological analysis revealed a relevant benefit of the combination strategy.

Treatment with PA-C combined with MSC and iPSC-NSC significantly preserved β-III-Tubulin positive fibers and synaptic buttons and reduced the glial scar area. Notably, PA-C treatment alone also significantly reduced the scar area and increased both the preservation of neurons and the number of synaptic buttons on intact neuronal somas surrounding the injured area compared to control, indicating a significant contribution of PA-C to the neuroprotective effect found in the combinatorial approach.

Previous studies in an Alzheimer’s disease model also demonstrated an increase in functional synapses following curcumin administration [69,70,71]. In this study, we believe that PA-C supported a similar neuroprotective effect after SCI by preventing axonal degeneration and maintaining the synaptic integrity of the neuronal circuits. PA-C may also contribute to the prevention of synaptic dysfunction and degeneration and the formation of novel synaptic connections.

Interestingly, iPSC-NSC or MSC treatment increased motoneuron preservation (in agreement with a previous study [72]), although combined iPSC-NSC and MSC treatment failed to provide significant preservation levels. While a previous study employing ex vivo spinal cord slices demonstrated how curcumin reduced motoneuron apoptosis [73], PA-C-treatment failed to protect motoneurons. As PA-C treatment increased the preservation of NeuN-positive cells (as previously demonstrated [74,75]), the neuroprotective effect may be attributed to a specific subset of neurons, as previously shown by Seo et al. in a model of Alzheimer’s disease, which cannot be identified in the quantification performed here using non-specific neural cell fate identification [76].

We established a significant reduction in demyelination, which occurs due to the progressive oligodendrocyte cell death after SCI [77], following treatment with either iPSC-NSC or MSC, but not PA-C. Previous studies found that NSC [78] and iPSC-NSC [79] undergo oligodendrocytic differentiation and remyelinate axons in vivo. Thus, transplanted iPSC-NSC may contribute to the remyelination by differentiating into oligodendrocytes, although MSC may induce endogenous oligodendrocytic differentiation by secreting factors such as neurotrophins. Previous studies describing the transplantation of NSC and bone-marrow MSC in SCI models support this hypothesis [80,81].

Prolonged inflammation and a lack of injury resolution represent some of the significant impediments to functional regeneration after SCI [82]. Hence, we investigated whether the neuroprotective effects following our various treatment strategies may derive from reduced astrogliosis [83] or altered microglia polarization [84]. After SCI, astrocytes undergo cellular, molecular, and functional changes; they also contribute to the release of inhibitory extracellular matrix components that impair axonal regrowth [85]. While we found that stem cell transplantation had no significant inhibitory effect on glial scar formation (indeed, iPSC-NSC may have contributed to an increase in glial scar area) (Figure 6A), PA-C treatment alone or in combination with iPSC-NSC and MSC transplantation significantly reduced the glial scar area following SCI.

Several signal transduction pathways are involved in astrogliosis, including STAT and NF-κB [86,87,88], and a recent study demonstrated that curcumin might reduce astrogliosis and glial scar formation after SCI through the inhibition of STAT3 [89] and NF-κB [90] pathways. Therefore, the anti-inflammatory activity of PA-C may contribute to a reduction in glial reactivity through the inhibition of NF-κB signaling, as shown in our in vitro models.

Histological analysis revealed that PA-C treatment or PA-C treatment in combination with iPSC-NSC and MSC transplantation prompted an increase in Arg1 expression levels, which is an M2 microglia marker associated with an anti-inflammatory polarization profile [91], among the total microglial cells found within the injury site [92,93]. LPS stimulation in vitro induces M1 phenotype polarization; however, curcumin can reverse this polarization and increase the expression of M2-associated biomarkers through the TLR4– NF-κB signaling pathway [94]. These previous studies agree with our in vitro LPS experiment in which PA-C reduced the translocation and activation of NF-κB. Thus, PA-C may drive microglial polarization by modulating the NF-κB pathway. While MSC also drive M2 polarization after acute SCI [30], the transplantation of iPSC-NSC and MSC failed to increase M2-polarized microglia post-SCI, with only the addition of PA-C affecting microglia polarization.

The complex pathological nature of SCI requires the implementation of a multifaceted and versatile therapeutic perspective regarding the development of treatments. A combined treatment comprising PA-C, iPSC-NSC, and MSC provides immunomodulatory and neuroprotective effects to prevent axonal degeneration, neuronal death, and loss of neuronal connectivity. Furthermore, this combinatorial strategy reduced the injury area and prevented the expansion of the glial scar in chronic stages, thereby providing a versatile and clinically relevant approach to treating acute SCI.

## 4. Materials and Methods

### 4.1. Human iPSC-NSC and MSC Culture Conditions

Human iPSC-NSC were generated as previously described [20]. Briefly, human iPSC were generated using synthetic mRNA transfection of CYCLIN D1 plus base reprogramming factors (OCT3/4, SOX2, KLF4, LIN28), which results in a significantly improved genetically stable footprint in human iPSC. This strategy enables the more accurate and reliable generation of human iPSC for disease modeling and future clinical applications. Ectoderm-like differentiation for iPSC-NSC generation used the PSC Neural Induction Medium kit (A1647801, Life Technologies, Carlsbad, CA, USA) and Geltrex-coated plates (A1413201, Life Technologies). After neural induction, iPSC-NSC were maintained in growth medium: STEMdiff™ Neural Progenitor Medium (05835, STEMCELL™, Vancouver, BC, Canada) supplemented with 200 U/mL penicillin (09-757F, LONZA, Basel, Switzerland) and 200 μg/mL streptomycin (09-757F, LONZA) in Geltrex-coated plates in standard cell incubation conditions. For iPSC-NSC expansion and sub-culture, cells were detached by enzymatic digestion with TrypLE^TM^ Select (12563-011, Thermo Fisher Scientific, Waltham, MA, USA) when cultures reached 80% confluence.

Human adipose-derived MSC were obtained from surplus suprapatellar fat as previously described [95] and ≈ 6500 cells/cm^2^ were cultured at 37 °C in High Glucose DMEM (SH30022.01, Cytiva HyCloneTM, Marlborough, MA, USA) supplemented with 10% fetal bovine serum (F7524, Sigma-Aldrich, Darmstadt, Germany), 2 mM L-glutamine (H3BE17-605E, LONZA), 0.3 g/mL glucose (G7021, Sigma-Aldrich), 200 U/mL penicillin, and 200 μg/mL streptomycin. MSC were dissociated with trypsin after reaching 80% confluence and expanded for up to ten passages.

### 4.2. Analysis of Cell Viability, Differentiation, and Neurite Outgrowth in iPSC-NSC

iPSC-NSC were characterized for neural feature markers after seeding on Geltrex-coated coverslips at a cell density of 20,000 cells per well and allowed to expand for 24 h. The expression of neural development markers was analyzed using conventional immunocytochemical staining protocol as described in Section 4.4.

PA-C toxicity and iPSC-NSC tolerance evaluated employed a dose–response experiment, and the MTS (3-(4,5-dimethyl-thiazol-2-yl)-5-(3-carboxymethoxy-phenyl)-2-(4-sulfo-phenyl)-2H-tetrazolium) was used to evaluate mitochondrial activity as an indirect measure of cell metabolic activity and thus used as a cell viability assay following the manufacturer’s instructions (ab197010, Abcam plc, Cambridge, UK). iPSC-NSC seeded at a density of 1.5 × 10^4^ cells/well on a Geltrex-coated 96-well plate with growth medium were allowed to attach and expand for 24 h and then treated with increasing concentrations of PA-C (3.8 *w*/*w* [37]), 5, 10, 12.5, 15, 17.5, and 20 µM for 24 h. Optical density was measured at 490 nm using a microplate reader Victor 2 (Perkin Elmer Inc., Waltham, MA, USA).

The effect of PA-C and C (B3347, TCI Europe, Zwijndrecht, Belgium) on neurite elongation in iPSC-NSC were also examined. Fas (F-4660, LC Laboratories, Woburn, MA, USA) and PGA-SS-Fas (Patent n° WO/2020/193802) were also used as positive controls for neurite elongation [96]. iPSC-NSC were seeded at a density of 6 × 10^4^ cells/well on Geltrex-coated coverslips 24 h before treatment with 10 µM C or PA-C and 50 µM Fas or PGA-SS-Fas. Neurite retraction was induced by adding 10 µM of LPA (L726, Sigma-Aldrich) and incubating the cultures for an additional 24 h. Neurite outgrowth was quantified from at least six fields (10×) of three independent experiments from β-III-Tubulin positive cells using the NeuriteJ plug-in from ImageJ as previously described [97], and the average neurite length is represented.

Neuronal differentiation was evaluated by immunostaining and PCR, using the RT2 Profiler PCR Array for human neurogenesis (PAHS-404ZA, Qiagen, Hilden, Germany). iPSC-NSC were seeded on Geltrex coated coverslips in P24 plates at a density of 6 × 10^4^ cells/well in growth medium for immunostaining and in P6 plates at a density of 6 × 10^5^ cells/well in growth medium for RNA extraction. Cells were incubated overnight before treatment with 10 µM PA-C or vehicle (PBS) for an additional 24 h.

The capacity of PA-C to prevent oxidative damage was evaluated using a toxic concentration of hydrogen peroxide to partially mimic the hostile environment generated at the injury site after SCI. NSC were seeded at a density of 1.5 × 10^4^ cells/well on a Geltrex-coated 96-well plate in growth medium and allowed to attach and expand for 24 h. Then, iPSC-NSC were treated with increased concentrations of PA-C (5, 10, 12.5, 15, 17.5, and 20 µM) for 30 min before exposure to 75 or 100 µM of H_2_O_2_ and incubated for an additional 24 h before MTS was performed. To elucidate the neuroprotective effect of PA-C on the survival of neural lineages, iPSC-NSC were seeded on Geltrex-coated coverslips in P24 plates at a density of 6 × 10^4^ cells/well with growth medium and allowed to attach and expand for 24 h and then pre-treated with 10 µM PA-C for 30 min before exposure to 75 µM of H_2_O_2_ and incubated for an additional 24 h.

### 4.3. NF-κB Activation Assay in iPSC-NSC + MSC Co-Cultures Treated with PA-C

To assess the potential immunomodulatory properties of PA-C, iPSC-NSC were incubated alone on Geltrex-coated coverslips in P24 plates at a density of 6 × 10^4^ or in co-culture with 2 × 10^4^ MSC, seeded on Matrigel-coated ThinCert cell culture inserts (662640, Greiner Bio-One, Madrid, Spain) with 0.4 µm of pore size at a density of 2 × 10^4^ in growth medium. Cell cultures were treated with 10 µM PA-C or vehicle for 24 h and then stimulated with 1 µg/mL LPS (L8274, Sigma-Aldrich) for an additional 24 h to activate NF-κB and nuclear translocation was evaluated by NF-κB immunostaining.

### 4.4. Immunocytochemistry

Cultured cells were fixed with 4% paraformaldehyde (PFA) for 15 min, washed three times with phosphate buffer solution, and blocked with 5% normal goat serum (Thermo Fisher) and 0.1% Triton X-100 (9036-19-5, Merck Millipore, Darmstadt, Germany). Cells were incubated with primary antibodies overnight at 4 °C in a humidified chamber. The primary antibodies used were mouse anti-β-III-Tubulin (1:400; MO15013, Neuromics, Edina, MN, USA), chicken anti-GFAP (1:1000; PA1-10004,Thermo Fisher), mouse anti-NFκB (p65) (1:400; sc-8008, Santa Cruz, Dallas, TX, USA), guinea pig anti-DCX (1:400; ab5910, Chemicon, Temecula, CA, United States), anti-Pax-6 (PAX6) (1:400; PRB-278P, Biolegend, San Diego, CA, USA), rabbit anti-SOX2 (1:400; MAB5326, Abcam), mouse anti-Nestin (1:400; MAB5326, Sigma Aldrich), mouse anti-Notch-1 (1:400; AF1057, R&D System, Minneapolis, MN, USA), mouse anti-O4 (1:200; MAB345, Sigma-Aldrich), and mouse anti-FOXJ1 (1:300; 14-9965-82, Thermo Fisher). For secondary antibodies, AlexaFluor 488, 555, and 647 (Invitrogen, Carlsbad, CA, USA) conjugated with respective IgGs were used at a dilution of 1:400 and incubated for 1 h at room temperature. Samples were counterstained with DAPI (1:1000) for 5 min and finally mounted using FluorSaveTM reagent (EMD Millipore, Burlington, MA, USA). Immunofluorescence images were captured from in vitro experiments using Zeiss ApoTome microscope (Carl Zeiss) and analyzed using ImageJ.

### 4.5. RNA Extraction, cDNA Synthesis, and RT2 Profiler PCR Human Neurogenesis Array

For transcriptomic analyses, total RNA was extracted using TRIzol^®^ reagent (15596026, Thermo Fisher Scientific) according to the manufacturer’s instructions. RNA concentrations were determined with a NanoDrop spectrophotometer (ND1000, NanoDrop Technologies, Wilmington, DE, USA) and reverse transcribed using the high-capacity RNA-to-cDNA™ kit (4368814, Applied Biosystems, Foster City, CA, USA) with 2.5 µg of total RNA, in a reaction volume of 20 µL, through incubation at 37 °C during 120 min using random hexamer primers.

PCR was performed using the RT2 Profiler PCR Array for human neurogenesis (PAHS-404ZA, Qiagen, Hilden, Germany). Following the manufacturer’s instructions, we used 27.17 ng of cDNA for quantitative PCR in a total volume of 25 µL using AceQ SYBR qPCR Master Mix (Q111-02, VazSyme Biotech Co., Nanjing, China) on a LightCycler 480 Instrument (Roche, Basel, Switzerland). The reaction sequence included 10 min of incubation at 95 °C, followed by 40 cycles of 20 s at 95 °C, 20 s at 60 °C, and 20 s at 72 °C.

The manufacturer’s template was used for gene expression analysis of genes involved in the neural cell fate determination, differentiation, and apoptosis. Relative gene expression levels were calculated using the ΔΔCt method, and the results were expressed with reference to the arithmetic mean of three reference housekeeping genes (Actin Beta (ACTB), Glyceraldehyde-3-Phosphate Dehydrogenase (GAPDH), and Hypoxanthine Phosphoribosyltransferase 1 (HPRT1)). Fold change was calculated by normalizing the expression rates of the treatment to the control sample. Genes with aberrant melting curves were discarded. Real-time PCR arrays were performed from at least three independent in vitro experiments. Paired Student’s *t*-tests were used to compare differences in gene expression between control iPSC-NSC and PA-C-treated iPSC-NSC.

### 4.6. Contusive Spinal Cord Injury, Cell Transplantation, and PA-C Delivery In Vivo

Female Sprague–Dawley rats weighing ≈300g were housed under controlled temperature under a 12 h light/dark cycle with ad libitum access to food and water. Animals were treated with morphine (2.5 mg/kg b.w s.c.) 30 min before surgery. Deep anesthesia was induced with 3% isoflurane and then maintained at 1.5–2% during surgery. Laminectomies were conducted at thoracic segments T7–T9 to perform a moderate contusion at T8 by applying 200 kdyn in all animals (n = 72) using the Infinite Horizon Spinal Cord Impactor (Precision Systems and Instrumentation, LLC) as previously described [98]. Animals were randomly distributed into the following groups: control (intramedullary injection of cell vehicle cell culture medium; and intrathecal administration of saline 0.9%), iPSC-NSC (intramedullary injection), MSC (intramedullary injection), iPSC-NSC + MSC (intramedullary injection of both iPSC-NSC and MSC), PA-C (intrathecal administration of PA-C), and iPSC-NSC + MSC + PA-C (intramedullary injection of both iPSC-NSC and MSC combined with intrathecal administration of PA-C). Stem cell transplants and/or PA-C treatment were performed seven days post-SCI. iPSC-NSC and/or MSC were collected and suspended in 14 µL of DMEM/F12 (11320033, Gibco, Grand Island, NY, USA) at a concentration of 130,000 cells/µL, or mixed 1:1 for co-transplantation, for a total of 1.8 × 10^6^ cells per animal. Control and PA-C treated animals received culture medium injections (DMEM/F12-14 µL) instead. Cell suspensions or vehicle were intramedullary injected at the epicenter of the lesion (6 µL), and 2 mm rostral (4 µL) and caudal (4 µL) was administered to the lesion site at a rate of 2 μL/min using a siliconized pulled glass needle (100 µm internal diameter) coupled to a 10 µL Hamilton syringe controlled by an automatic injector. Immediately after transplantation, a catheter connected to an osmotic pump (model 1007D, Alzet^®^, Cupertino, CA, USA) was introduced into the intrathecal space from the fifth lumbar segment to guarantee sustained local delivery of PA-C. The pump delivered 1 µL/h for seven days and was filled with 100 µL of 4 mM PA-C diluted in 0.9% saline. After seven days, pumps and catheters were removed.

As previously described, all animals were subjected to post-surgery care, including passive and active rehabilitation protocols [98]. To prevent immune rejection of the allogeneic cell grafts, animals received daily subcutaneous injections of the immunosuppressant FK506 (1 mg/kg) starting one day before transplantation, which was maintained for one month.

### 4.7. Functional In Vivo Locomotion Analysis

Locomotor recovery was evaluated using the open-field BBB locomotor scale [48] and the video-based system for automated gait analysis CatWalk^®^ (Noldus, Asheville, NC, USA). Animals were individually videotaped for 4 min, and two unbiased observers blindly scored the results. CatWalk analysis was performed in the ninth week. Paw contact was quantified by counting high-intensity pixels as the mean of at least three rounds per analysis [99].

### 4.8. Histological Studies of Spinal Cords

At week nine after SCI, animals were irreversibly anesthetized by intraperitoneal injection of sodium pentobarbital (100 mg/kg) and fentanyl (0.05 mg/kg) and transcardially perfused with 0.9% saline immediately followed 4% PFA in 0.1 M phosphate buffer (pH 7.4). Spinal cords were dissected and post-fixed in 4% PFA for 5 h and then conserved in 0.1 M phosphate buffer containing 0.01% sodium azide. Thoracic segments, including T6 to T10, were either cryopreserved, immersed in 30% sucrose before inclusion in Tissue-Teck OCT (Sakura Finetek Europe BV, Flemingweg, Netherlands), and stored at −80 °C until cryo-sectioning or dehydrated and included in paraffin, placed in histology cassettes, and processed on a Leica ASP 300 tissue processor (Leica Microsystems, Nussloch, Germany). Then, 8 µm thick longitudinal sections in the horizontal plane were cut on a cryostat or microtome and mounted on gelatin-coated slides, with six series collected.

One complete series was employed to perform hematoxylin/eosin (H/E) staining using an automated station (Autostainer XL Leica, Leica Biosystems, Wetzlar, Germany), scanned on the Aperio Versa scanner (Leica Biosystems), and analyzed using the Aperio ImageScope software (Leica Biosystems). H/E staining was used to determine the anatomical structure and quantify preserved motoneurons nine weeks post-SCI. Spinal cord topography previously described by Waibl [100] was employed to identify the epicenter of the lesion. The center of the T8 thoracic segment was identified as the lesion epicenter, with T7 as the rostral site to the lesion and T9 as the caudal site to the lesion. T7 and T9 were further subdivided into 1 mm segments (as shown in Figure 6A). Motoneurons were quantified considering previously described criteria: polygonal cells with a diameter of ≥20 µm located at the ventral horns (extensively reviewed at [101]). The absolute number of motoneurons was determined along the thoracic segments T7, T8, and T9, measured every 1 mm, and normalized to the area analyzed (data expressed as MN/mm^2^).

The analysis of the amount of spared white matter and the quantification of myelin preservation used LFB (IW-3005, IHC World, LLC, Ellicott City, MD, USA) staining on longitudinal spinal cord sections. Spinal preparations were scanned on the Aperio Versa scanner (Leica Biosystems). For analysis, images were adjusted to threshold by color, saturation, and brightness to select the signal determined by myelin axons in the white matter and analyzed using the “analyze particles’’ function of ImageJ. The signal-positive area was normalized to the total area of spinal cord tissue.

The glial scar area was analyzed by quantifying the lack of GFAP staining and expressed as the percentage of the total spinal cord area analyzed (establishing a fixed area analyzed in all samples; 8 mm length). For the quantification of preserved neuronal β-III-Tubulin fibers, the background signal was first subtracted; then, images were adjusted to threshold by color, saturation, and brightness to select the specific signal and analyzed using the “analyze particles’’ function of ImageJ. The signal-positive area was normalized to the total area of spinal cord tissue analyzed.

To evaluate the functional synapsis, the co-localization of Synaptophysin and NeuN-positive signal (NeuN+/Synaptophysin+) was quantified at the epicenter of the injury analyzed using the “analyze particles’’ function of ImageJ. The positive NeuN positive area was used to estimate the preservation of neurons in the injured spinal cord.

To assess the grade of microglia polarization at the injury area, double staining of IBA1 (a marker of microglia) and Arg1 (an anti-inflammatory marker) (IBA1+/Arg1+) was quantified using the “analyze particles’’ function of ImageJ and positive area was normalized to the total area of spinal cord tissue analyzed.

For the immunohistochemistry analysis, background signal was first subtracted, and images were adjusted to a threshold by color, saturation, and brightness to select the specific signal prior to each analysis.

The long-term survival of iPSC-NSC and MSC in the spinal cord was evaluated at week nine by immune detection of human mitochondria for iPSC-NSC and detection of human mitochondria and the specific mesenchymal cell maker CD-105 for MSC (Appendix A).

Immunofluorescence for paraffin-embedded spinal cord sections required prior dewaxing, rehydration, and antigen retrieval (immersion in tris-EDTA buffer (10 mM Tris, pH 9.0) for 25 min at 97 °C) steps. Longitudinal sections were incubated with blocking solution (5% horse serum, 10% fetal bovine serum in phosphate buffer solution with 0.1% Triton X-100) for 1 h at room temperature and incubated with primary antibodies overnight in a humidified chamber. The following primary antibodies and the indicated dilutions were used: mouse anti-β-III-Tubulin (1:400; MO15013, Neuromics), rabbit anti-GFAP (1:1000; PA1-10004, Dako Denmark A/S, Glostrub, Denmark), chicken anti-NeuN (1:400; ABN91, EMD Millipore), rabbit anti-IBA-1 (1:500; 019-19741, Wako Chemicals USA Inc., Bellwood, VA, USA), mouse anti-Arg1 1 (1:400; sc-271430, Santa Cruz), mouse anti-synaptophysin (1:400; sc-17750, Santa Cruz), mouse anti-human mitochondria (1:400; MAB1273, Chemicon International Inc., Temecula, CA, USA), and anti-CD-105 (1:200 ab53321, Abcam). AlexaFluor 488, 555, and 647 (1:400; Invitrogen) conjugated with secondary antibodies against the respective IgG were used. We stained nuclei with DAPI (1:1000) and mounted samples using FluorSaveTM Reagent (EMD Millipore). H/E, LFB, and immunofluorescence sections were scanned using the Aperio Versa scanner (Leica Biosystems) and analyzed using the Aperio ImageScope software (Leica Biosystems) or ImageJ.

### 4.9. Ethical Statement

All animal experiments were undertaken in accordance with guidelines established by the European Communities Council Directive (210/63/EU) and the Spanish regulation 1201/2005. All experimental procedures were approved by the Animal Care and Use Committee of the Research Institute Prince Felipe (2021/VSC/PEA/0032) and the Ethics Committee in Animal and Human Experimentation of the Universitat Autònoma de Barcelona (procedure #10121). All animals were managed by professionally trained staff.

The generation of iPSC and their implementation in this study were approved by the Dirección General de Investigación y Alta Inspección Sanitaria (Generalitat Valenciana, Valencia, Spain) with reference number 4/2020.

Human adipose tissue was obtained from surplus fat tissue during knee prosthesis surgery under sterile conditions. The human samples were anonymized, and the experimental procedure was previously evaluated and accepted by the Regional Ethics Committee for Clinical Research with Medicines and Health Products following the Code of Practice 2014/01. As exclusion criteria, no samples were collected from patients with a history of cancer or infectious diseases at the time of the surgery (viral or bacterial). All human patients voluntarily signed an informed consent document for the use of the adipose samples.

### 4.10. Statistical Analyses

Statistical analyses were performed using Graph Pad Prism Software (GraphPad Software, San Diego, CA, USA). Outliers were identified and removed by the ROUT method provided by Graph Pad Prism Software, and normality was assessed (Shapiro-Wilk, Kolmogorov–Smirnov, D’Agostino and Pearson, and Anderson–Darling tests) before performing the corresponding statistical analyses. *p*-value less than 0.05 was considered significant, and was indicated as followed: * *p* < 0.05; ** *p* < 0.01; *** *p* < 0.001; **** *p* < 0.0001.

## Figures and Tables

**Figure 1 ijms-22-05966-f001:**
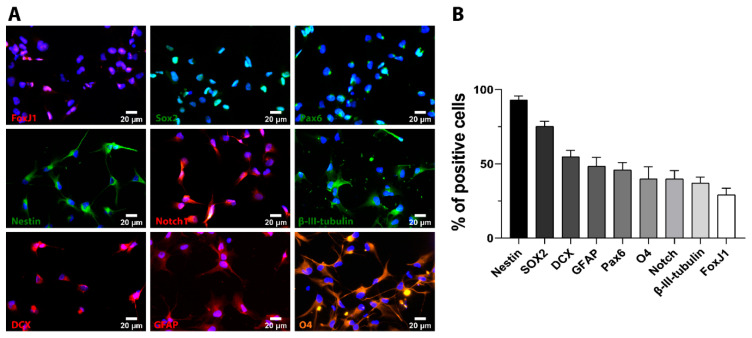
In vitro characterization of cultured iPSC-NSC. (**A**). Representative immunofluorescence staining images for the indicated cell markers. DAPI (blue) employed for nuclei counterstaining. Scale bar = 20 µm. (**B**). Quantitative analysis of the percentage of positive cells for each cell marker. Data expressed as mean ± S.E.M. from three independent cell culture experiments.

**Figure 2 ijms-22-05966-f002:**
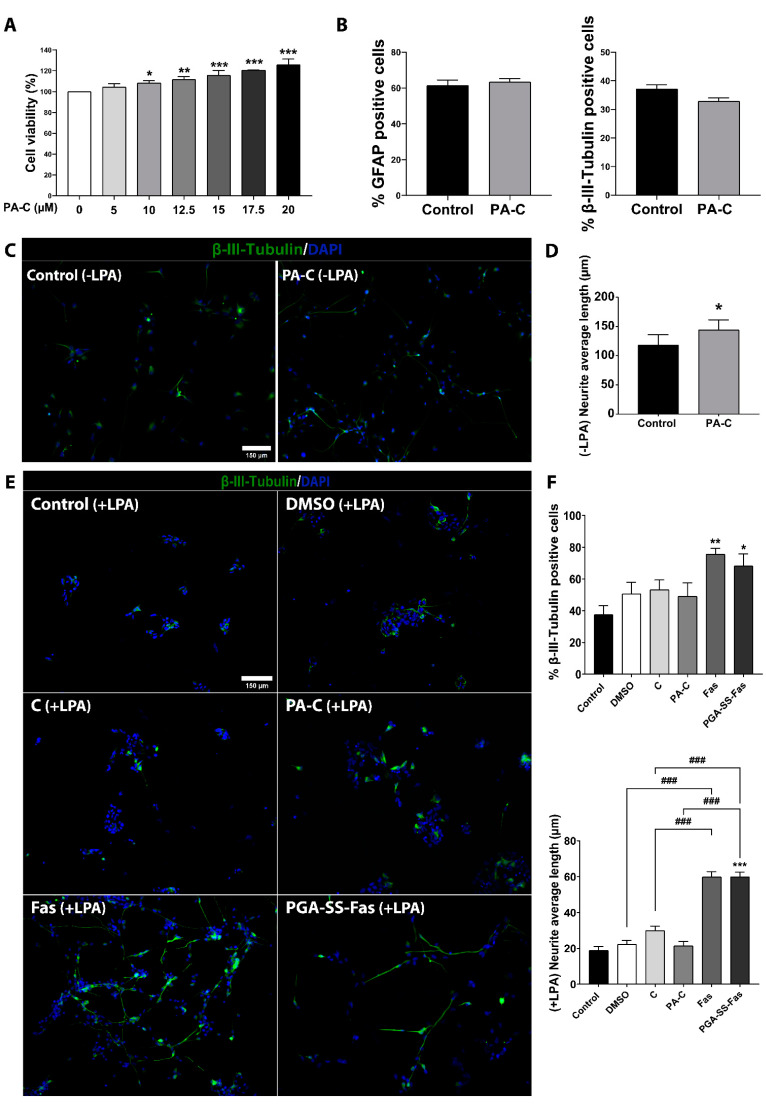
(**A**). Effect of PA-C treatment on iPSC-NSC viability and neurite outgrowth in vitro. (**A**). Cell viability evaluations of iPSC-NSC treated with increasing concentrations (5, 10, 12.5, 15, 17.5, and 20 µM) of PA-C for 24 h. Values represented as a percentage relative to vehicle-treated control iPSC-NSC. Data expressed as mean ± S.E.M. of three independent experiments determined by one-way ANOVA with Dunnett’s multiple comparison test (* *p* < 0.05, ** *p* < 0.01, *** *p* < 0.001 vs. control). (**B**). Immunofluorescence quantification of the percentage of iPSC-NSC positive for astroglial (GFAP; left panel) and neuronal (β-III-Tubulin; right panel) markers after treatment with 10 µM PA-C for 24 h. (**C**). Representative images of immunofluorescence staining of β-III-Tubulin (green) and nuclear DAPI staining (blue) (scale bar = 150 µM). (**D**). Quantitative analysis of neurite elongation in iPSC-NSC treated with 10 µM PA-C for 24 h. Data represented as average neurite length (µm) and expressed as mean ± S.E.M. of three independent experiments determined by unpaired Student’s t-test (* *p* < 0.05 vs. Control). (**E**). Representative images of immunofluorescence staining of β-III-Tubulin (green) and nuclear DAPI staining (blue) (scale bar = 150 µm). (**F**). Quantitative analysis of neurite elongation in iPSC-NSC pre-treated with 10 µM PA-C or 50 µM PGA-SS-Fas for 24 h and then treated with LPA for a further 24 h. Data represented as average neurite length (µm). Data expressed as mean ± S.E.M. of three independent experiments determined by one-way ANOVA with Tukey’s multiple comparison test and unpaired Student’s *t*-test (* *p* < 0.05, ** *p* < 0.01, *** *p* < 0.001 vs. Control; ### *p* < 0.001 as indicated).

**Figure 3 ijms-22-05966-f003:**
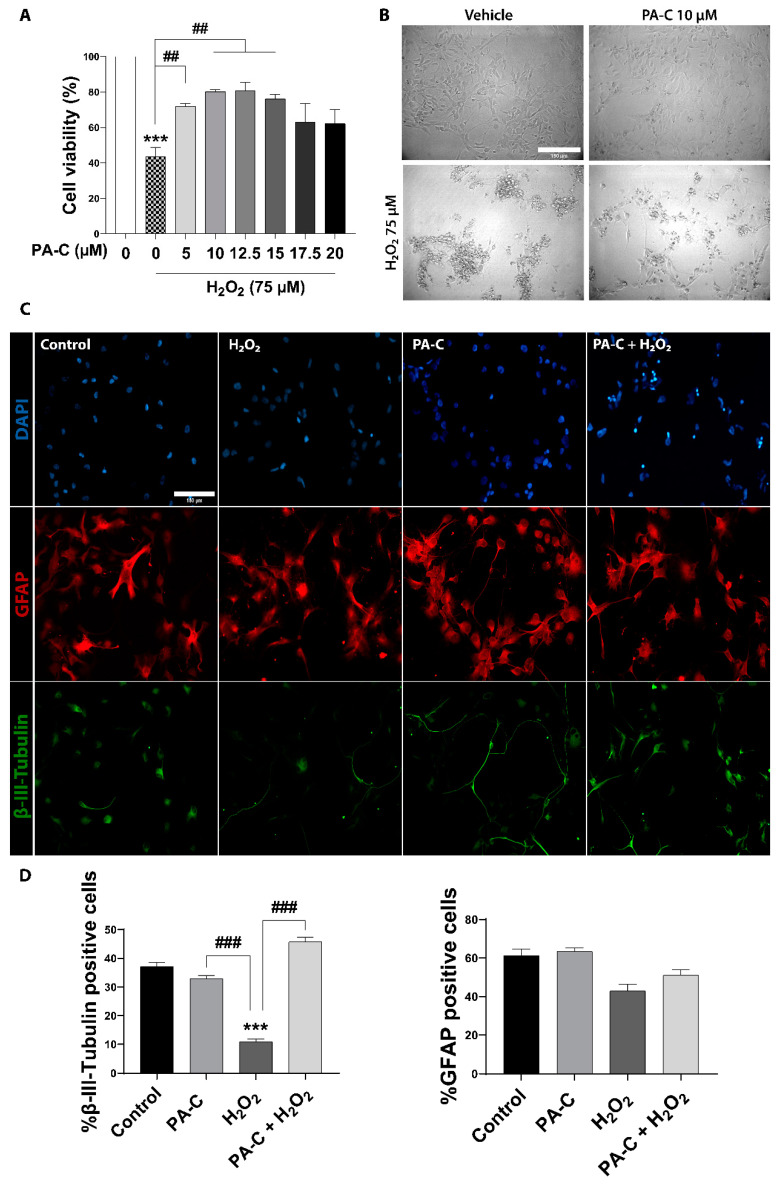
Neuroprotective effect of PA-C pre-treatment following peroxide-induced cytotoxicity in iPSC-NSC. (**A**). Effect of PA-C pre-treatment (5, 10, 12.5, 15, 17.5, and 20 µM) on iPSC-NSC viability following a 24 h incubation with 75 µM H_2_O_2_. MTS viability values represented as a percentage of vehicle-treated control cells. Data expressed as mean ± S.E.M. of three independent experiments determined by one-way ANOVA multiple comparison tests (*** *p* < 0.0001 vs. Control; ## *p* < 0.01 as indicated). (**B**). Representative bright-field images of iPSC-NSC morphology after PA-C pre-treatment and H_2_O_2_ treatment (scale bar = 150 µm). (**C**). Representative immunofluorescence staining of β-III-Tubulin (green), GFAP (red), and DAPI (blue) of iPSC-NSC pre-treated for 30 min with 10 µM PA-C and then incubated with 75 µM H_2_O_2_ for 24 h. Scale bar = 150 µm. (**D**). Quantitative analysis of the ratio of β-III-Tubulin cells and GFAP cells shown as a percentage of total DAPI from immunostaining. Data expressed as mean ± S.E.M. from three independent experiments as determined by one-way ANOVA with Tukey’s multiple comparison test (*** *p* < 0.001 vs. Control; ### *p* < 0.001 as indicated).

**Figure 4 ijms-22-05966-f004:**
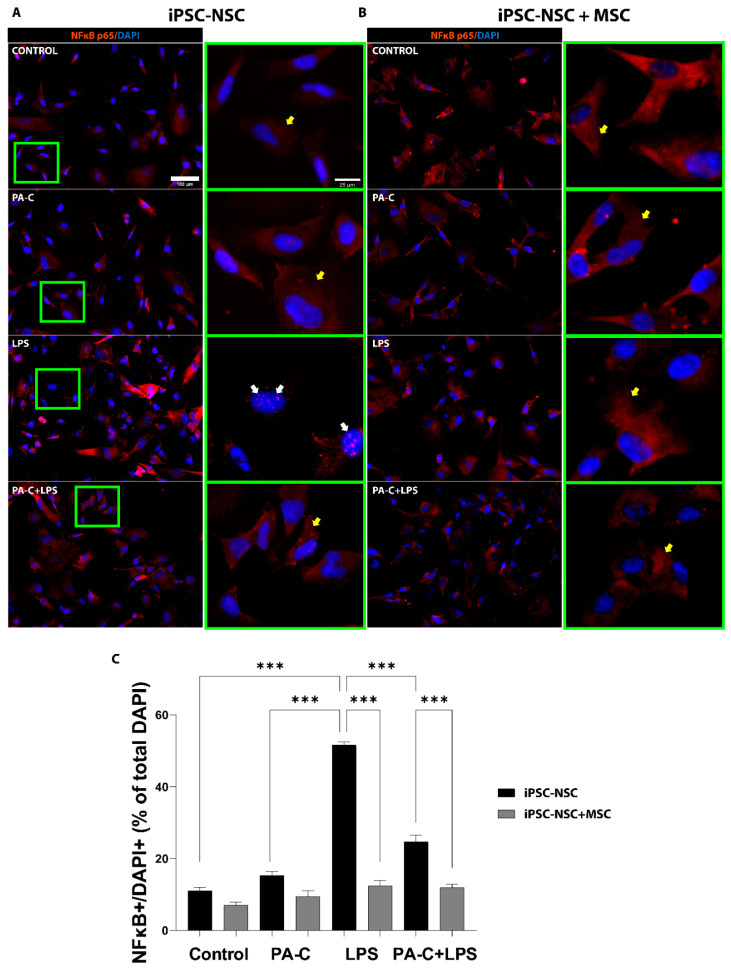
PA-C pre-treatment and MSC co-culture prevent NF-κB translocation/activation in iPSC-NSC following pro-inflammatory LPS insult. (**A**,**B**). Representative immunostaining images of the NF-κB inflammatory marker (orange) and nuclear staining DAPI (blue) in iPSC-NSC monoculture (iPSC-NSC) (**A**) and MSC co-culture (iPSC-NSC + MSC) (**B**) incubated for 24 h with 10 µM PA-C or vehicle and then incubated with LPS (1 µg/mL) or vehicle for 24 h (scale bar = 100 µm). Green squares show higher magnification of the nuclei (scale bar = 20 µm), with white and yellow arrows indicating nuclear and cytosolic NF-κB, respectively. (**C**). Quantification of NF-κB translocation by quantifying iPSC-NSC with nuclear NF-κB (NF-κB+/DAPI+) expressed as a percentage of the total number of iPSC-NSC by DAPI. Data expressed as mean ± S.E.M. (at least eight random fields from three independent experiments) determined by one-way ANOVA with Tukey’s multiple comparison test (*** *p* < 0.001 as indicated).

**Figure 5 ijms-22-05966-f005:**
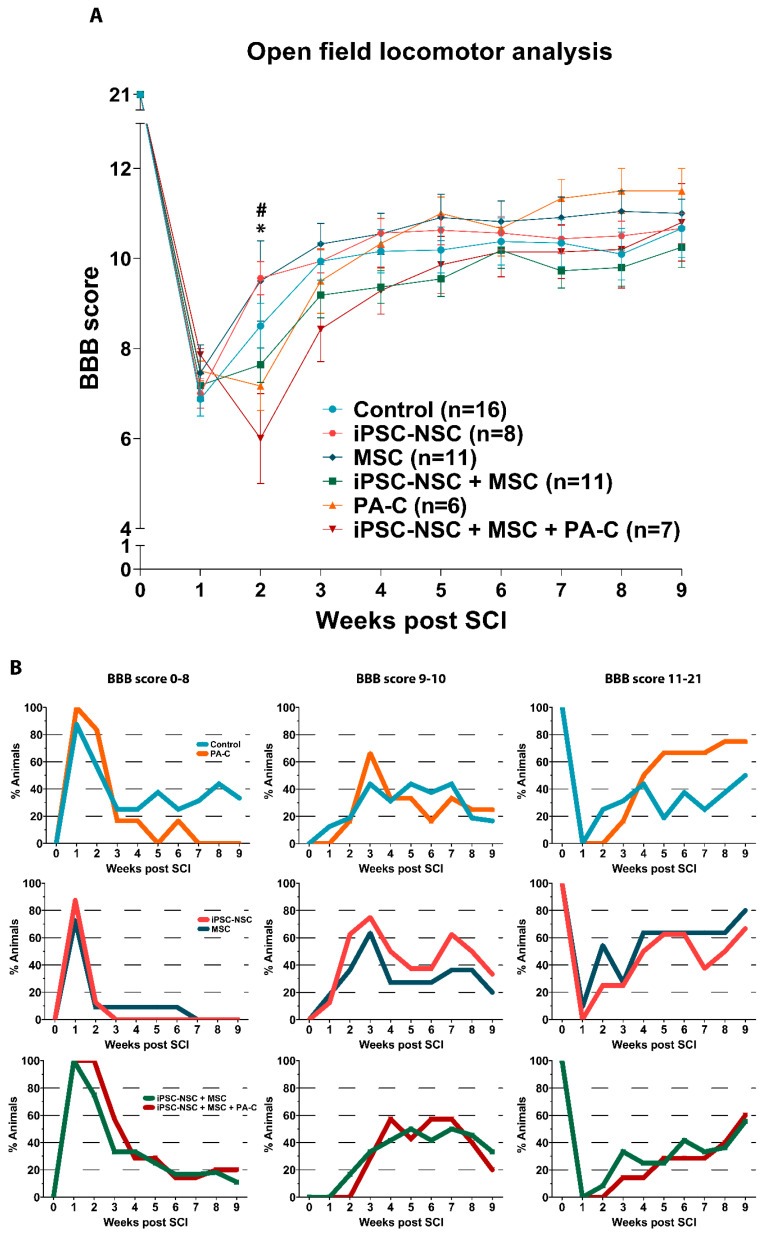
Locomotor recovery after SCI following single treatments and combination therapy. (**A**). Time course locomotor evaluation by open-field BBB scale over nine weeks post-SCI. (**B**). BBB locomotor evaluation subdivisions (0–8, 9–10, and 11–21) showing the percentage of animals in each score range. Treatments compared at all post-injury time points. Data expressed as mean ± S.E.M. (control n = 16; iPSC-NSC n = 8; MSC n = 11; iPSC-NSC + MSC n = 11; PA-C n = 6; iPSC-NSC + MSC + PA-C n = 7) determined by two-way mixed model ANOVA with Tukey’s multiple comparison test (* *p* < 0.05 iPSC-NSC vs. PA-C; # *p* < 0.01 iPSC-NSC vs. iNSC + MSC).

**Figure 6 ijms-22-05966-f006:**
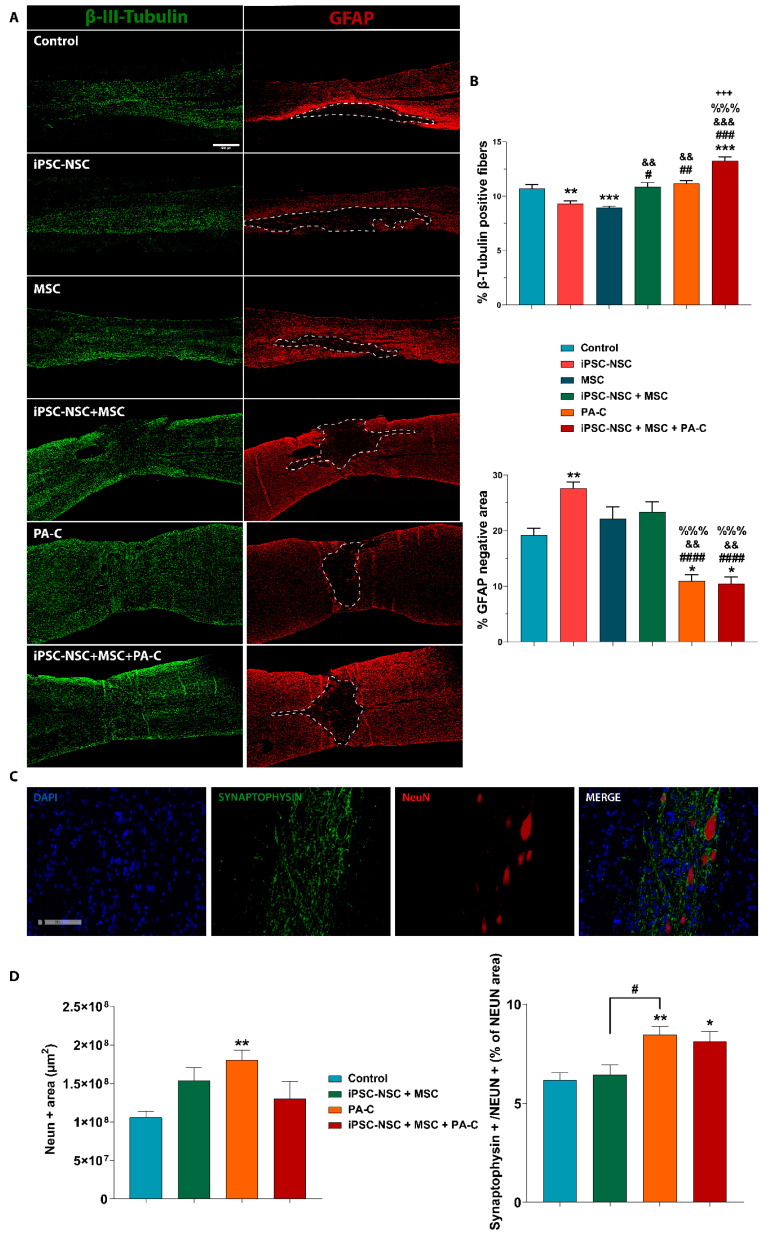
(**A**). PA-C and PA-C combined with iPSC-NSC + MSC transplantation preserves β-III-Tubulin fibers and synapses and reduces scar size A. Representative immunofluorescence images (**left panel**) of β-III-Tubulin (green) and GFAP (red) of longitudinal spinal cord sections, including the injured area nine weeks after SCI (scale bar = 500 µm). Dotted lines delimit the GFAP-negative area. (**B**). Quantification of β-III-Tubulin-positive fibers (**upper panel**) and GFAP-negative scar area (**lower panel**) represented as a percentage of the total analyzed area and expressed as mean ± S.E.M. determined by one-way ANOVA with Tukey’s multiple comparison test for β-III-Tubulin and Kruskal–Wallis one-way ANOVA with Dunn’s method for GFAP (control, n = 8; iPSC-NSC + MSC, n = 8; n = 4 for iPSC-NSC, MSC, PA-C, iPSC-NSC + MSC + PA-C). * *p* < 0.05, ** *p* < 0.01, *** *p* < 0.001 vs. control; # *p* < 0.05, ## *p* < 0.01, ### *p* < 0.001, #### *p* < 0.0001 vs. iPSC-NSC; && *p* < 0.01, &&& *p* < 0.001, vs. MSC, %%% *p* < 0.001 vs. iPSC-NSC + MSC, +++ *p* < 0.001 vs. PA-C. (**C**). Representative immunofluorescence images of longitudinal spinal cord sections of synaptophysin (green), NeuN (red), and DAPI (blue) staining (scale bar = 200 µ). (**D**). Quantification of NeuN-positive area (**left panel**) and functional synapses by analyzing the co-localization of synaptophysin and NeuN (**right panel**), represented as a percentage of the NeuN-positive area. Quantitative data expressed as mean ± S.E.M determined by one-way ANOVA with Tukey’s multiple comparison test (Control n = 4; iPSC-NSC + MSC n = 3; PA-C n = 3; iPSC-NSC + MSC + PA-C n = 3) (* *p* < 0.05, ** *p* < 0.01 vs. control; # *p* < 0.05 as indicated).

**Figure 7 ijms-22-05966-f007:**
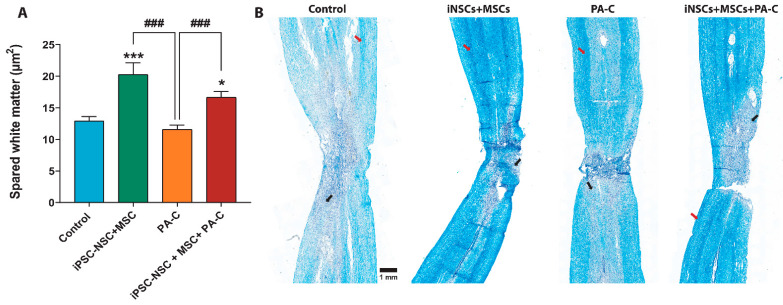
PA-C combined with iPSC-NSC + MSC transplantation increases white matter sparing. (**A**). White matter sparing analysis expressed as mean ± S.E.M determined by one-way ANOVA with Tukey’s multiple comparison test (n = 3). * *p* < 0.05, *** *p* < 0.001 vs. control; ### *p* < 0.001 as indicated. (**B**). Representative images of spinal cord longitudinal sections (right panel; scale bar = 1 mm); Black arrows and red arrows indicate demyelinated areas and myelinated areas, respectively.

**Figure 8 ijms-22-05966-f008:**
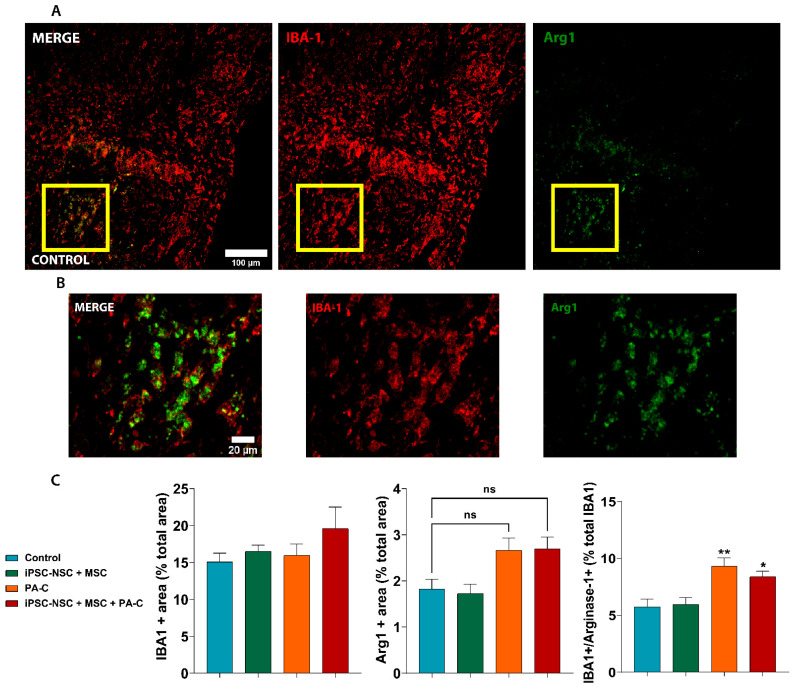
PA-C and PA-C combined with iPSC-NSC + MSC transplantation prompt microglial polarization towards an anti-inflammatory profile. (**A**). Representative immunostaining images of IBA1 (microglia marker; red) and Arg1 (anti-inflammatory marker; green) (scale bar = 100 µm). (**B**). Magnified images of the indicated area in A with a yellow square for IBA1 and Arg1 staining (scale bar = 20 µm) (**C**). Quantification of total microglia, positive for IBA1 (left panel) and the total number of cells positive for Arg1 (central panel), which are represented as the percentage of positive cells of the tissue area analyzed; Right panel represents the percentage of activated microglia, co-expressing IBA1 and Arg1, represented as a percentage of the total microglia (positive for IBA1). Quantitative data expressed as mean ± S.E.M determined by one-way ANOVA with Tukey’s multiple comparison test (n = 3). * *p* < 0.05, ** *p* < 0.01 vs. control; ns (non-significant) as indicated.

## Data Availability

No new data were created or analyzed in this study. Data sharing is not applicable to this article.

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
