# Peer review of "Human-Induced Neural and Mesenchymal Stem Cell Therapy Combined with a Curcumin Nanoconjugate as a Spinal Cord Injury Treatment"

_ijms, 2021, doi:10.3390/ijms22115966_

Round 1
Reviewer 1 Report
- “From this select group of genes, only Dopamine receptor D2 (DRD2) expression became significantly downregulated after iNSC treatment with PA-C (compared to control vehicle-treated iNSC), indicating that PA-C treatment did not significantly induce neural differentiation of iNSC.”
- No figure is addressed and more details of the exact genes should be provided.
- “Overall, these findings provide evidence for the anti-inflammatory effect of PA-C pre-treatment and MSC co-culture in iNSC cultured under pro-inflammatory conditions.”
- Inflammatory is not the best term that can be used for in vitro experiment that do not involve any immune cells. It is recommended to be changed throughout the manyscript.
- Line- 290 “-III Tubulin-positive area than iNSC or MSC treatments.” The font should be checked.
- The dotted line pattern is confusing and should be changed in Figure
- Figure 6C. It is not clear to what type of treatment the figure is related.
- Figure 7 Regions of interest should be pointed or arrows.
- The dotted line pattern is confusing and should be changed in Figure S4B.
- The dotted line pattern is confusing and should be changed in Figure 8C.
- Some capital letters are missing in the Materials and Methods section.
Reviewer 2 Report
General comments
In this study, Pablo Bonilla and colleagues report the therapeutic efficacy of the combined delivery of human iPSC-derived neural stem cells (iPSC-NSC), human mesenchymal stem cells (MSC), and a polyacetal-curcumin nanoconjugate (PA-C), when applied to a rat model of spinal cord injury (SCI). Previous studies from the same group had shown benefits from combining the transplantation of rat neural progenitors with the in situ delivery of PA-C, (in terms of locomotor function, neuronal preservation, and reduced scarring). The novelty of the present study relies on adding human MSC to the previous combinatorial approach and on using a clinically relevant source of NSC (iPSC-NSC). The methodologies chosen, which include a clinically-relevant rat model of SCI (contusion) are adequate, the treatment/analysis of the resultant data was overall appropriate (see specific comments below), conclusions are well supported by results, and the manuscript well written.
In vitro studies showed that PA-C promotes iPSC-NSC cell viability and neurite elongation, as also reported by the authors for rat NSC, besides protecting iPSC-NSC from neuroinflammation and from hydrogen peroxide-induced toxicity. In vivo studies showed that compared to single treatments, this combination therapy is more effective in reducing the inhibitory scar area and increasing neuronal fiber preservation following SCI, though these improvements did not translate into locomotor functional recovery. These findings go beyond the state of art of regenerative therapies for treatment of SCI, and, as such, are scientifically relevant.
Nevertheless, some issues need to be addressed/clarified, as detailed below:
Specific comments:
1) The term iNSC is most often associated with NSC obtained by direct conversion (reprogramming) of adult somatic cells, such as human skin fibroblasts or human peripheral blood cells (see for instance the paper from Oliver Brüstle’s group https://doi.org/10.1038/s41467-018-06398-5) as well as this https://doi.org/10.1186/s40478-020-00960-3). I advise the authors to use an alternative abbreviation e.g. iPSC-NSC.
2) The following sentence is too long, and difficult to understand – “ The now US Food and Drug Administration (FDA)-approved immortalized NSC line (NSI-566) derived from human early fetal spinal cord tissue recently underwent feasibility and safety evaluations to treat chronic SCI in phase I clinical trials, reporting no serious adverse effects [9]”.
3) The benefits of MSC transplantation are associated primarily to their paracrine effects, as stated by authors in the following sentence – “The therapeutic benefits of MSC primarily relate to the secretion of paracrine acting factors, ….” As such, the use of MSC in this study instead of concentrated MSC secretome (or MSC-derived extracellular vesicles) should be justified. Alternatively, authors should address in discussion the potential interest of using MSC secretome in combinatorial therapies involving NSC transplantation, instead of MSC.
4) NSC characterization by IC (Figure 1A and B) showed a high percentage of cells expressing the NSC marker nestin, as expected, but also a significant percentage of cells expressing neuronal/oligodendrocytic/astrocytic markers. Is this expected when using this neural induction protocol with iPSC? Please discuss this issue and provide adequate references.
5) Line 134 – The following statement “moreover, PA-C treatment significantly increased iNSC proliferation… (Figure 2A)” needs revision since MTS provides cell metabolic activity data. To infer cell proliferation from MTS data, a standard curve (obtained plotting Ab values against known cell numbers of the same cell suspension) would be needed. Without a standard curve, one can only state that the increase in cell metabolic activity may be associated with increased cell numbers. Revise results and discussion accordingly.
6) In Figures 2A and 2F (upper figure) the conditions statistically different from the control condition were highlighted with asterisks. If authors were only interested in detecting differences from a single condition (control), please justify the use of the Tukey’s Pos hoc test for multiple comparisons instead of the Dunnett's test.
7) In Line 171 (Figure 2 legend) note that Beta III tubulin is a neuronal marker and not a neural marker. Correct accordingly.
8) In Figure 3A, according to the graph, the sole condition leading to reduced cell viability when compared to control is cell exposure to H2O2 without PA-C treatment. Treatment with PA-C resulted in cell viability levels similar to the control, independently of the PA-C concentration. Please confirm.
9) In Lines 228-229 - “Additionally, PA-C pre-treatment significantly reduced LPS-induced NF-κB translocation in iNSC monocultures or iNSC+MSC co-culture”. Which is the evidence supporting that PA-C pre-treatment reduces LPS-induced NF-κB translocation in iNSC+MSC co-cultures?
10) Line 557 – the expression “… to activate NF-κB activation” needs editing.
11) MSC exert immunomodulatory, anti-inflammatory, neurotrophic/neuroprotective and pro-angiogenic effects on the host microenvironment. Did the authors checked the integrity of the blood spinal cord barrier at the injured site apart from analyzing microglia polarization, white matter sparing, glial scar size, and neuronal fiber preservation?
12) Locomotor function data was analyzed using 2-way ANOVA. Since the measurements repeatedly taken from the same animals, a mixed model repeated measures ANOVA (in which time is the within-subjects factor and condition the between-subjects factor) would better allow the detection of a significant effect imparted by conditions.
13) Line 265 – Authors report that“…more than 80% of animals treated with iNSC, MSC, or PA-C reached a BBB range of 11-21” (Figure 5B), but I could not find evidences supporting this statement in Figure 5B for animals receiving only iNSC or PA-C treatment.
14) The legends of figures depicting IHC results (Figures 6, 7 and 8) should include the range of animals used for the correspondent quantitative image analysis data shown, instead of reporting the minimal number of animals used, as this was low (n = 3). Also, take note that in figure 7 legend, the number of animals used to quantify white matter sparring (Figure 6A) is missing.
15) In Discussion (line 369), in the sentence - “Finally, Sun et al. demonstrated that the sub-acute transplantation of human NSC and MSC enhanced cell survival, ...”, specify which cell type contributed to survival enhancement“.
16) In methods, authors should refer the rationale for the selection of the sample size in in vivo experiments (refer for instance if a priori power analysis was performed).
